# The Tuning Capability of CuO and Na₂CO₃ Dopant on Physical Properties for Laser Sealing Using Fiber Types of Sealant

**So Young Kim [1,2]**, **June Park [1]**, **Seon Hoon Kim [1]**, **Linganna Kadathala [1]**, **Jong Hyeob Baek [1]**, **Jin Hyeok Kim [2]** and **Ju Hyeon Choi [1,\*]**

[1] Korea Photonics Technology Institute, 9, Cheomdan Venture-ro 108beon-gil, Buk-gu, Gwangju 61007, Korea; siong8100@gmail.com (S.Y.K.); jpark@kopti.re.kr (J.P.); shkim@kopti.re.kr (S.H.K.); lingannasvu@kopti.re.kr (L.K.); jhbaek@kopti.re.kr (J.H.B.)

[2] Department of Materials Science, Chonnam National University, 77, Yongbong-ro, Buk-gu, Gwangju 61186, Korea; jinhyeok@chonnam.ac.kr

[\*] Correspondence: juchoi2@kopti.re.kr; Tel.: +82-62-605-9265

**Abstract:** PbO-SiO₂-Al₂O₃-B₂O₃ (PSAB)-based glasses were prepared in order to determine the feasibility for laser sealing in the form of fiber. To reach a high quality of laser sealing, the tuning capability of CuO and Na₂CO₃ dopant concentration was examined on thermal, thermo-mechanical, and optical properties of glasses. The difference of thermal expansion coefficient was reduced with codoping of 1 wt% CuO-2 wt% Na₂CO₃ into the PSAB glass system, and it amounted to $0.34 \times 10^{-6}$/K. The codoped PSAB glass system reached 100% absorption at 810 nm, which corresponds to the wavelength of laser. The glass fiber with a diameter of 180 μm was successfully pulled from the codoped PSAB glass system. The fluorine-doped tin oxide (FTO) glass substrate using the glass fiber was successfully sealed. It presented a crack-free sealed surface with a sealing strength of about 41 MPa. These results indicate that the PSAB-based glass system in the form of fiber is proved as a laser sealing material in packaging systems.

**Keywords:** lead silicate glass; glass transition temperature; coefficient of thermal expansion; glass fiber; laser sealing

## 1. Introduction

State-of-the-art technology of industrial lasers has seen laser sealing techniques become an increasingly popular alternative for reliable packaging in many fields of manufacturing [1,2]. Laser sealing techniques have been intensively researched in packaging systems, such as dye sensitized solar cells, Organic Light-Emitting Diode (OLED, and solid oxide fuel cells [3–8]. These systems used the conventional furnace-based and laser-assisted glass frit sealing processes until now. However, these processes cause the durability, efficiency, and life of the packaging systems to deteriorate. The laser sealing process has the disadvantage that the pores are present after sealing, reducing the durability. Thus, less pores, uniform height of sealing preform, 100% hermetic sealing, and low cost processing are essential for industrial applications [9]. In this direction, laser sealing using glass fibers has been attractive owing to reducing the defective rate by improving the ability of sealant thickness. It also controls the shape to prevent internal defects formation, such as pore and gas, during the process.

On the other hand, in order to minimize the damage of the device, previous reports suggested that sealing glass material should have a low glass transition temperature and thereby low thermal energy transfers to devices [9,10]. Additionally, the coefficient of thermal expansion (CTE) of sealing glass material should be similar to that of the panel. The CTE difference between the glass panel and

laser sealing glass should be lower than $1.0 \times 10^{-6}$/K [11]. If the CTE is not matched, stress will be generated between the two materials, causing deformation and breakage of the material in all kinds of packaging applications [10–15]. A few studies were carried out on the tuning of CTE between laser sealing glass and glass panels. The CTE of $Bi_2O_3$-$B_2O_3$-ZnO-$SiO_2$-based glass systems was found to be in the order of $10.0 \times 10^{-6}$/K for solar cell panels.

In this work, for the first time, to the best of our knowledge, we investigated the thermal, thermo-mechanical, and optical properties of the PbO-$SiO_2$-$Al_2O_3$-$B_2O_3$ (PSAB) glass system doped with CuO and $Na_2CO_3$ for laser sealing in the packaging systems. We studied the role of CuO and $Na_2CO_3$ as dopants in this system with two functions that control the CTE between the panels and increase the absorption of the 810 nm band that corresponds to the laser wavelength. The optimized CuO and $Na_2CO_3$ codoped PSAB glass system was also pulled into a glass fiber. The fluorine-doped tin oxide (FTO) glass substrates were sealed using glass fiber by controlling the laser power, laser scan velocity, and laser scan cycle. Finally, the optimized sealing condition was evaluated by photography of sealed fiber and tensile strength.

## 2. Materials and Methods

The PSAB glasses with composition of PbO-$SiO_2$-$Al_2O_3$-$B_2O_3$-xCuO-y$Na_2CO_3$ (x = 0.5, 1, 1.5, and 2 wt%; y = 0.5, 1, 3, and 5 wt%) were prepared by the melt-quenching method. Reagent grade chemicals of PbO (99.9%), $SiO_2$ (99.9%), $Al_2O_3$ (99.99%), $B_2O_3$ (99.9%), CuO (99.9%), and $Na_2CO_3$ (99.9%) purchased from Alfa-Aesar were used as raw materials to fabricate the glasses. The batch chemicals (about 50 g) were weighed using a microbalance and then crushed using a ball mill with zirconia balls for 30 min to get a homogeneous mixture. Then, the well-mixed powder was transferred into an alumina crucible and heated in an electric furnace at 750 °C for 60 min under $N_2$ atmosphere. After one hour, the melt was cast onto a preheated brass mold and then transferred to another electric furnace to anneal the glass samples. These glasses were annealed at 350 °C for 3 h to get thermal stability in the glasses. The optimized PSAB glass system codoped with CuO and $Na_2CO_3$ was pulled into a fiber with 180 μm diameter using a draw tower in the temperature range of 500–535 °C.

The amorphous nature of glasses was examined by means of X-ray diffractometer (X'pert Pro, Panalytical) with CuK$\alpha$ (=1.542 Å) used as source. Simultaneous thermal analyzer (STA 409PC, NETZSCH) was used for thermal behavior of the glass samples. The measurement was carried out in the range of 25–900 °C at the heating rate of 10 °C/min under $N_2$ atmosphere. Thermo-mechanical analysis (TMA) was carried out using a dilatometer (DIL402F3, NETZSCH) in the range of 0–450 °C with a heating rate of 5 °C/min under $N_2$ atmosphere. The glass samples with a dimension of 5 (H) $\times$ 5 (W) $\times$ 8 (L) $mm^3$ were used for the measurement. The transmittance spectra were recorded with a UV-VIS-NIR spectrophotometer (Cary 500 scan, Varian H) in the range of 400–1100 nm with a spectral resolution of 1.0 nm. The glass powders were made into cylindrical pellets (diameter 10 mm and height 10 mm) by pressing. The flow button test was performed. The pellets on FTO substrates were sintered in the oven at 400–550 °C for 30 min. The laser sealing process was performed by a custom made laser sealer composed of a galvanometer (SG7210, Sinogalvo) and a CW diode laser with center wavelength of 810 nm. In the process, the laser beam with a spot diameter of 3 mm and maximum output power of 60 W focused to the FTO substrates placed on the hot plate. After being sealed with fiber types of sealant, the sealing strength tests were performed using a universal tensile tester (AG-X, Shimadzu) to evaluate the laser sealing quality. In this test, five measurements were taken, and the average value was given as a result.

## 3. Results and Discussion

Figure 1a depicts the DTA (differential thermal analysis) traces of PSAB glasses in dependence of CuO concentration increased from 0–2.0 wt%. The thermal properties, such as glass transition temperature ($T_g$) and crystallization temperature ($T_x$), were taken as illustrated in Figure 1a. It was noticed that the $T_g$ remained almost unchanged at around 387–389 °C regardless of CuO dopant

concentration. However, $T_x$ considerably increased up to 509 °C from 480 °C with the addition of 0.5 wt% of CuO to the base glass, and it remained constant for further increase in CuO concentration. The thermal stability ($\Delta T$) is frequently characterized by the difference between $T_g$ and $T_x$, $\Delta T = T_x - T_g$. Figure 1b shows the quantitative values of $T_g$, $T_x$, and $\Delta T$, dependent on CuO concentration. The $\Delta T$ increased compared to the base PSAB glass, and it remained almost constant regardless of CuO dopant concentration.

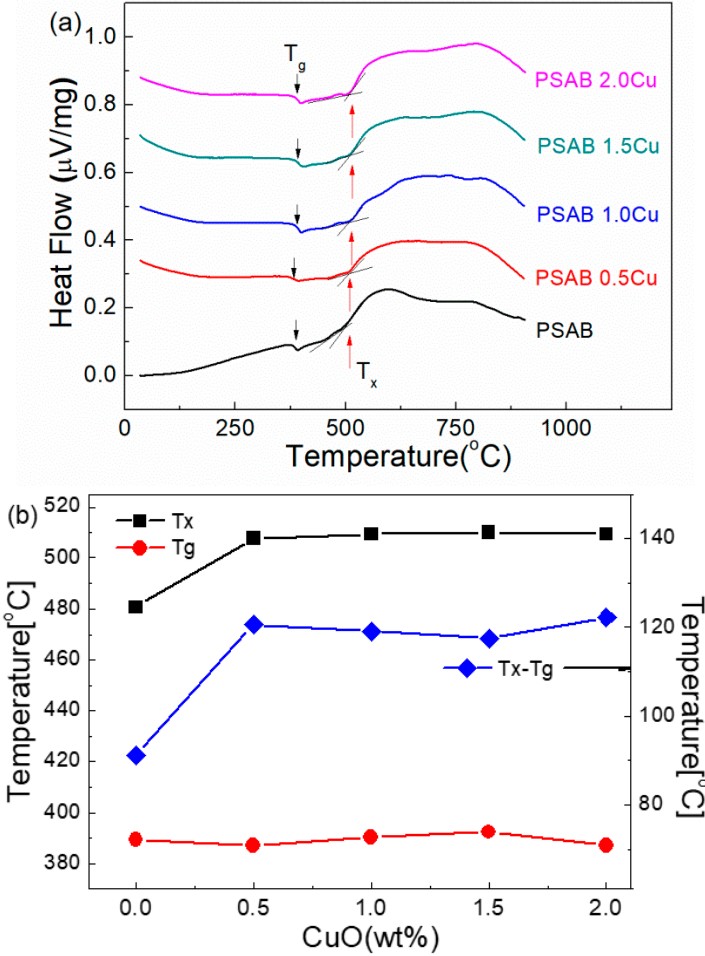

**Figure 1.** (**a**) Differential thermal analysis (DTA) traces; and (**b**) the quantitative values of glass transition temperature ($T_g$), crystallization temperature ($T_x$), and thermal stability $\Delta T$ ($T_x - T_g$) in a PbO-SiO$_2$-Al$_2$O$_3$-B$_2$O$_3$ (PSAB) glass system doped with various CuO concentrations.

The change in $T_g$ as a function of CuO concentration has been explained in other reported glasses [16–18]. On one hand, it was reported that the $T_g$ increased with increase in CuO concentration in P$_2$O$_5$-Na$_2$O-CuO glass network system. This was attributed to the increased formation of P-O-Cu bonds, resulting in an increased cross-linking density in the P$_2$O$_5$-Na$_2$O-CuO glass network. Therefore, the thermal and mechanical properties of the glasses were enhanced. On the other hand, there were several reports of the opposite effect of CuO on $T_g$, i.e., a decrease in $T_g$. It was reported that the $T_g$ decreased as the CuO concentration increased in the SiO$_2$-Li$_2$O-Al$_2$O$_3$-CuO glass system [19]. Justyna Sułowska et al. presented that the $T_g$ decreased as the CuO concentration increased in SiO$_2$-P$_2$O$_5$-K$_2$O-MgO-CaO-CuO glasses. Because the more covalent character of Cu$^{2+}$-O bonds replaced the more ionic bonds, such as Ca-O bonds and Mg-O bonds, this caused the glass structure to be more rigid, which resulted in the increasing amount of stress in the glass and thus, the $T_g$ decreased [20]. H. Takebe et al. reported that the $T_g$, CTE, and molar volume decreased with increasing CuO concentration in the CuO-BaO-B$_2$O$_3$-P$_2$O$_5$ glass system [21]. This behavior, i.e.,

the decreasing trend of $T_g$ and CTE simultaneously was not well explained. Therefore, a more systematic investigation of $T_g$ and CTE according to CuO concentration was carried out using the TMA method, which uncovered microscopic trends on the effect of CuO in glass systems.

Figure 2a shows the TMA traces of the PSAB glass system as a function of CuO dopant concentration. From TMA spectra, the CTE, $T_g$, and softening temperature ($T_s$), were obtained, dependent on CuO concentration. The inset of Figure 2a showed that the $T_g$ and $T_s$ decreased from 404 to 392 °C and 435 to 429 °C as CuO concentration increased from 0 wt% to 2.0 wt%, respectively. Figure 2b shows the quantitative values of CTE as a function of CuO dopant concentration for different temperature ranges. As CuO dopant concentration increases, the CTE decreases from $9.07 \times 10^{-6}$/K to $8.81 \times 10^{-6}$/K, $9.07 \times 10^{-6}$/K to $8.81 \times 10^{-6}$/K, and $8.65 \times 10^{-6}$/K to $8.17 \times 10^{-6}$/K for the temperature ranges of 25–100 °C, 25–200 °C, and 25–300 °C, respectively.

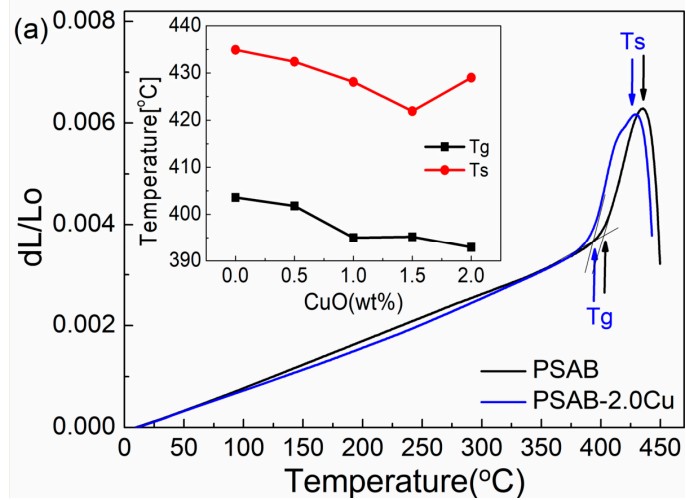

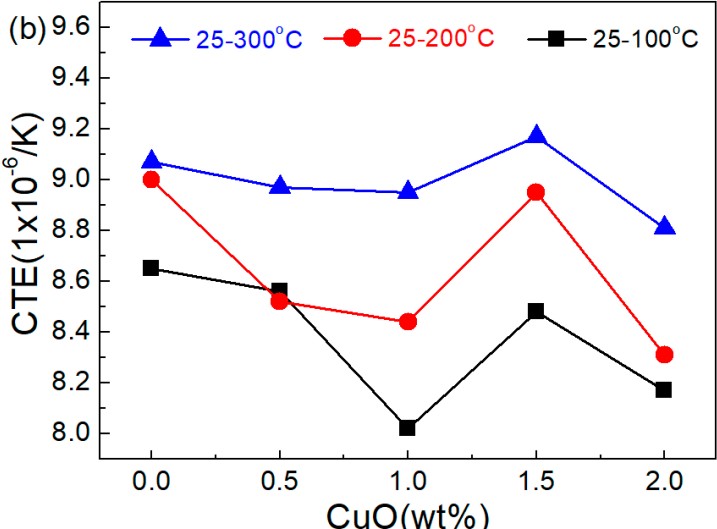

**Figure 2.** (**a**) Thermo-mechanical analysis (TMA) traces of the PSAB system doped with various CuO concentrations; and (**b**) the variation of coefficient of thermal expansion (CTE), dependent on CuO concentration for different temperature ranges; inset shows the variation of $T_g$ and $T_s$, dependent on CuO concentration.

The $T_g$ and $T_s$ are very important parameters in selecting glass materials for practical applications. The effect of CuO concentration on $T_g$ via Simultaneous Thermal Analysis (STA) was not clearly observed, as shown in Figure 1a. The TMA showed the decreasing trend of $T_g$, $T_s$, and CTE as a



function of CuO dopant concentration. The difference in the trend of $T_g$ from DTA and TMA is due to sensitivity variation of the techniques. In the DTA experiment, there is concern that the small amount material (few milligrams) used in the test may not be representative of the overall component. In the TMA experiment, the volume changes are measured in a material as a function of temperature, time, and an applied force. The effect of volume change is much more sensitive than the effect of specific heat change. In general, a decrease of $T_g$ is predicted by a decrease of rigidity in structure. The previous study reported the similar trend that the $T_g$ decreased as the CuO content increased [19]. It indicated that the introduction of CuO was assumed to convert the bridging oxygen bonds into nonbridging oxygen bonds in the lithium silicate glass network. Justyna Sułowska et al. [20] reported the similar behavior in $SiO_2$-$P_2O_5$-$K_2O$-$MgO$-$CaO$-$CuO$ glasses. Takebe et al. [21] reported that the $T_g$, CTE, and molar volume decreased with increasing CuO dopant concentration. The main reason for a decrease in $T_g$ was thought to be the converting of the bridging oxygen bonds into nonbridging oxygen bonds in the glass network as explained above. However, a decrease of CTE with the CuO addition into the PSAB glass system is to be studied more.

From Figure 2, it was found that the significant difference of CTE between 1 wt% CuO doped PSAB glass and the FTO coated glass panel was $2 \times 10^{-6}$/K. In order for the CTE difference to be below $1 \times 10^{-6}$/K, the optimization of composition is needed. It is basically well known that the addition of $Na_2O$ in the silicate network provides nonbridging oxygens and thus decreases the $T_g$ [22]. However, when the $Na_2O$ content was more than 6 wt%, it was reported that the chemical stability becomes deteriorated [23]. Therefore, we prepared a PSAB glass system doped with $Na_2CO_3$ concentration varied in the range of 0.5–5 wt% and studied its effect on $T_g$, $T_s$, and CTE.

Figure 3a shows the DTA traces of PSAB glasses, dependent on $Na_2CO_3$ concentration. Figure 3b shows the quantitative values of $T_g$ and $T_x$ as a function of $Na_2CO_3$ dopant concentration. As $Na_2CO_3$ dopant concentration increased up to 5 wt%, the $T_g$ and $T_x$ decreased from 380 to 360 °C, and 510 to 480 °C, respectively. The $\Delta T$ remained almost unchanged regardless of $Na_2CO_3$ dopant concentration. The higher the value of $\Delta T$, the greater the delay in the nucleation process, the greater the thermal stability, and the easier the glass formation [23]. A similar effect was observed in sodium silicate glasses and borate glasses when $Na_2O$ was added to the glass [22,24]. The constant of $\Delta T$ regardless of $Na_2CO_3$ dopant concentration indicated that there was no big difference in the nucleation process. Therefore, it was thought that the decrease of $T_g$ in this work was mainly due to the formation of nonbridging oxygens.

It was aforementioned that the CTE between 1 wt% CuO doped PSAB glass and the FTO coated glass panel was in the order of $2 \times 10^{-6}$/K. In order for the CTE difference to be below $1 \times 10^{-6}$/K, the PSAB glass system was codoped with 1.0 wt% CuO, and $Na_2CO_3$ concentration varied from 0 wt% to 2.5 wt%. Figure 4a shows the TMA profiles of codoped PSAB glass system as a function of $Na_2CO_3$ concentration. The inset of Figure 4a presents that increasing $Na_2CO_3$ concentration up to 2.5 wt% results in a decrease of $T_g$ from 395 to 375 °C. The CTE of the codoped PSAB glass system as a function of $Na_2CO_3$ dopant concentration for different temperature ranges is shown in Figure 4b. It was noticed that the CTE increased from $8.10 \times 10^{-6}$/K to $9.31 \times 10^{-6}$/K, $8.45 \times 10^{-6}$/K to $9.61 \times 10^{-6}$/K, and $8.90 \times 10^{-6}$/K to $9.89 \times 10^{-6}$/K for the temperature ranges of 25–100 °C, 25–200 °C, and 25–300 °C, respectively. In the case of the codoped PSAB system, the $T_g$ decreased and the CTE increased with increasing $Na_2CO_3$ dopant concentration. The difference of CTE was found to be low of the order of $0.34 \times 10^{-6}$/K between the FTO coated glass panel ($10.23 \times 10^{-6}$/K) and 1 wt% CuO-2 wt% $Na_2CO_3$ codoped PSAB glass system ($9.89 \times 10^{-6}$/K). The major reason for the decrease of CTE difference in the codoped PSAB glass system may be due to $Na_2CO_3$ concentration that was not attributed to the fact that the adding of $Na_2CO_3$ in the silicate network provides nonbridging oxygens and thus decreases the $T_g$, as explained in Figure 3.

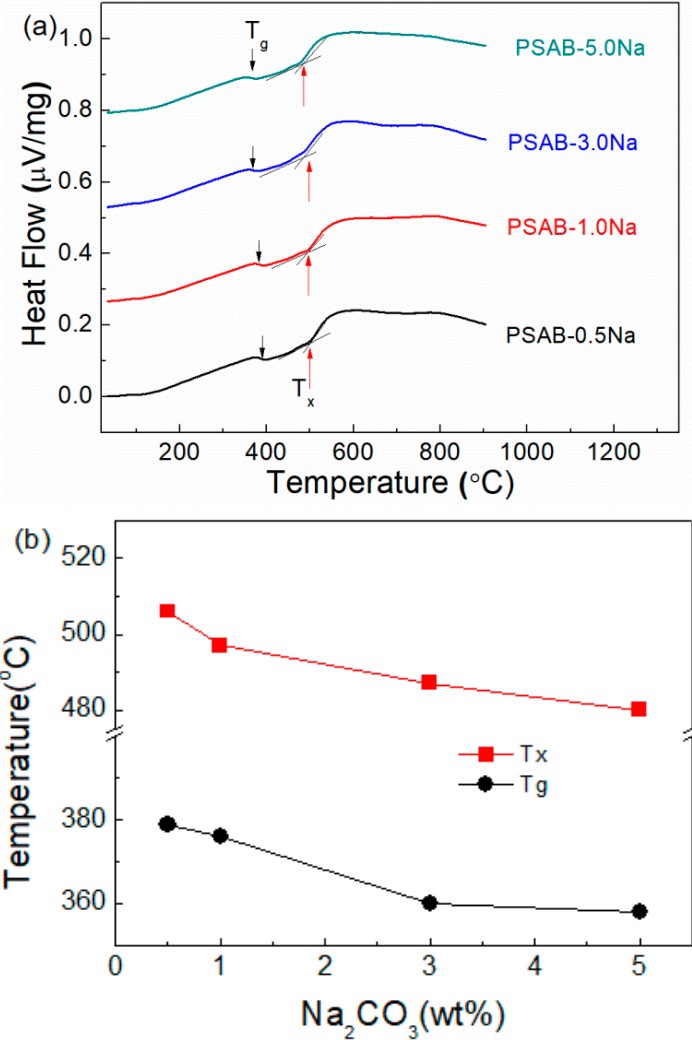

**Figure 3.** (**a**) The DTA traces; and (**b**) variation of $T_g$ and $T_x$ as a function of $Na_2CO_3$ concentration in the PSAB glass system.

In order to investigate the absorption at 810 nm, the transmission spectra of CuO-singly doped and CuO-Na$_2$CO$_3$ codoped PSAB glasses are shown in Figure 5. Figure 5a presents the transmittance spectra of CuO-doped PSAB glasses in the region 300–1100 nm for different concentrations of CuO. From the spectra, the bands at 500 and 810 nm are assigned to the Cu$^{2+}$ transitions of 3d$^{10}$ → 3d$^9$ 4s$^1$ and $^2$B$_{1g}$ → $^2$B$_{2g}$, respectively [25,26]. In the base PSAB glass system, the average transmittance between 500 and 1100 nm was about 70%. As CuO concentration increased, the transmittance decreased because of the UV absorption edge. The transmittance at 810 nm for 0.2 wt% and 0.5 wt% CuO-doped PSAB glasses was found to be 20.7% and 3.19%, respectively. For 1 wt% CuO, the transmittance was not observed at 810 nm. This result revealed that the absorption intensity and broadening at 810 nm increased with increasing CuO concentration. The broadening of this band may be ascribed to the superposition of three electron transition in 'd' orbitals corresponding to the $^2$B$_{1g}$ → $^2$E$_{g'}$ $^2$B$_{1g}$ → $^2$A$_{1g'}$ and $^2$B$_{1g}$ → $^2$B$_{2g}$ transitions of Cu$^{2+}$ ions.

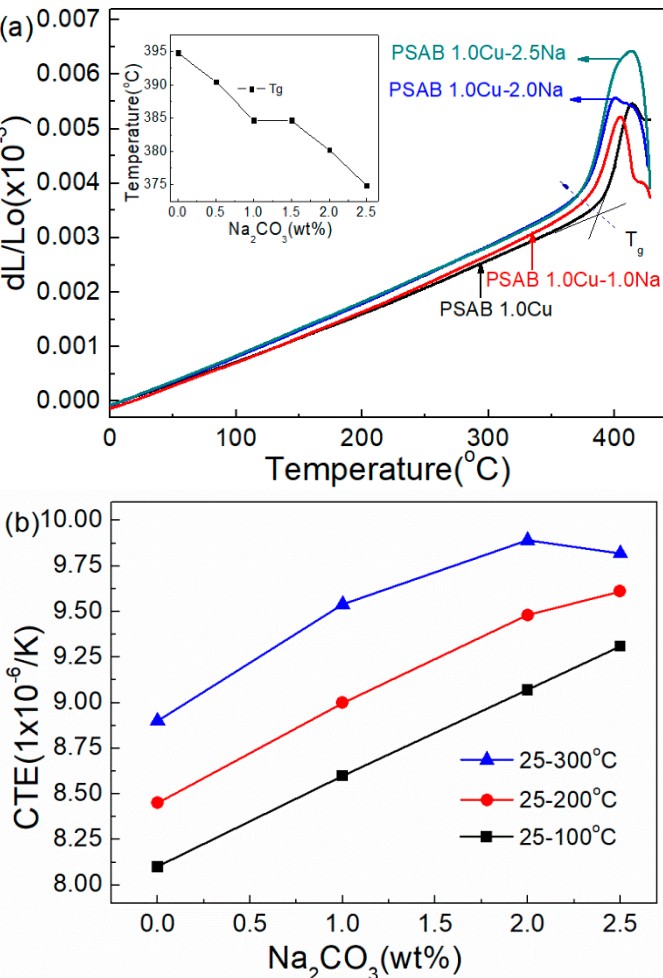

**Figure 4.** (**a**) TMA traces; and (**b**) variation of CTE for different temperature ranges, dependent on Na$_2$CO$_3$ dopant concentration; inset shows the variation of T$_g$, dependent on Na$_2$CO$_3$ concentration.

It was also observed that the absorption edge shifted towards higher wavelength, and the optical bandgap decreased with increasing CuO concentration. The reason for this behavior was explained by other researchers [27,28]. According to the report, the octahedral coordinated Cu$^{2+}$ ions act as modifiers to induce nonbridging oxygens (NBOs) in the glass network. Therefore, when CuO concentration increases, this resulted in an increase of NBOs. This leads to an increase in the degree of localization of electrons, thereby increasing the donor centers in the glass matrix. The presence of larger concentrations of these donor centers decreases the optical bandgap and shifts the absorption edge towards the higher wavelength side. In order to investigate the effect of CuO and Na$_2$CO$_3$ codopant on transmittance, the glass samples were prepared by adding 3 wt% Na$_2$CO$_3$ into 0.2 wt% and 0.5 wt% CuO-doped PSAB glass system. Figure 5b displays the transmittance of codoped PSAB glasses for different concentrations of CuO. It was noticed that the transmittance decreased from 20.7% to 13.4% and 3.19% to 0% at 810 nm with the addition of 3 wt% Na$_2$CO$_3$ into 0.2 wt% and 0.5 wt% CuO-doped PSAB glass system, respectively. It was also clear that the same trend was noticed when CuO was added into the glass system; however, the codoping into the system reached 100% absorption at the laser wavelength. With the addition of Na$_2$O alkali oxide into the glass system, a bond in the network was broken, resulting in an increase of NBOs and the relatively mobile Na ion became a part of the structure, which caused lower melting and working temperatures by decreasing viscosity [29]. The results indicate that codoping into the PSAB system not only increased the CTE but also decreased the transmittance, i.e., increased the absorption at 810 nm.

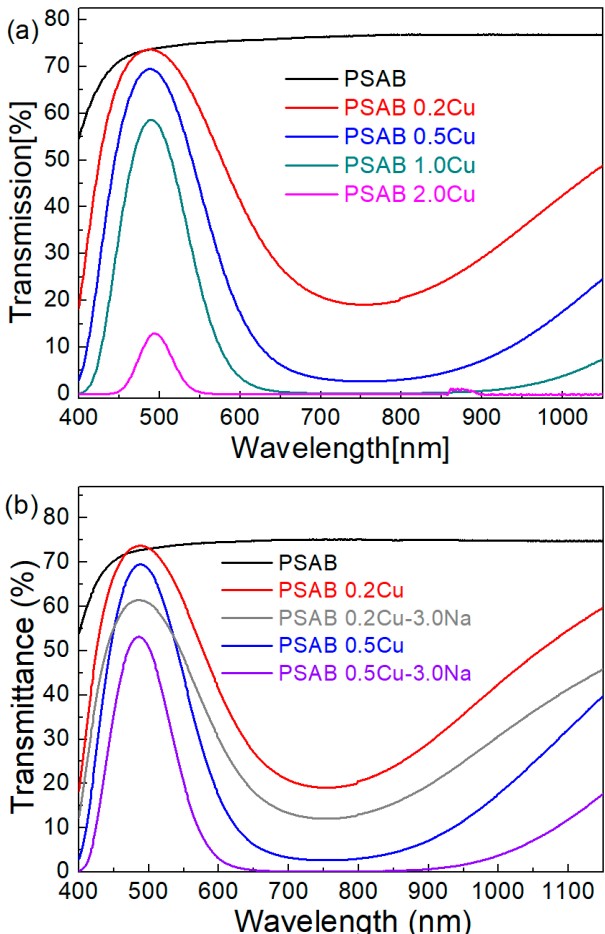

**Figure 5.** (**a**) Transmission spectra of PSAB system doped with various CuO concentrations; and (**b**) transmission spectra of PSAB system codoped with various CuO and $Na_2CO_3$ concentrations.

The flow button test was performed, in order to investigate flow ability of PSAB glass on substrate. The heat reaction of glass frits was divided into three stages [30,31]. In the first stage, the square shape changed to a trapezoid and then to a square. The pellet was eventually formed into a swollen sphere in the second stage. Thus, the shape of the pellet changed into an ellipsoid with an angle >90° at $T_s$. This is known as the softening point. In the third stage, the wetting angle of the molten glass was <90°. At higher temperatures (>$T_s$), the swollen sphere started to spread over and then the molten glass was eventually puddled on the substrate.

As shown in Figure 6, the flow button test showed the shapes of pellets as a function of sintering temperature at (a) 400 °C, (b) 450 °C, (c) 500 °C, and (d) 550 °C. Height and diameter of pellets were measured as a function of sintering temperature. The wetting angle of the pellet was divided into three stages as shown in Figure 6. At the first stage, a cylinder type of pellet was sintered at 400 °C for 30 min, and the volume of the pellet decreased by 23% to reduce the surface energy. The $T_g$ of PSAB glass frit was 390 °C, as shown in Figure 2. It could lead to a reduced volume of pellet due to the rearrangement and densification of the glass frits with square shape. At the second stage, the square types of pellet changed to swollen sphere after being sintered at 450 °C for 30 min. It indicated that flow ability is enough for laser sealing at this temperature, which is above the softening temperature of 430 °C as shown in Figure 2. In this stage, the wetting angle of the molten glass was >90°. At the third stage, the wetting angle of the molten glass started to show <90° at 500 °C. It was found that the molten glass spread over FTO coated glass due to viscous flow above 550 °C. Finally, it was confirmed that the flow ability was found to be good for laser sealing in the temperature range of 400–430 °C, which is above the softening temperature of 430 °C, as shown in Figure 2.

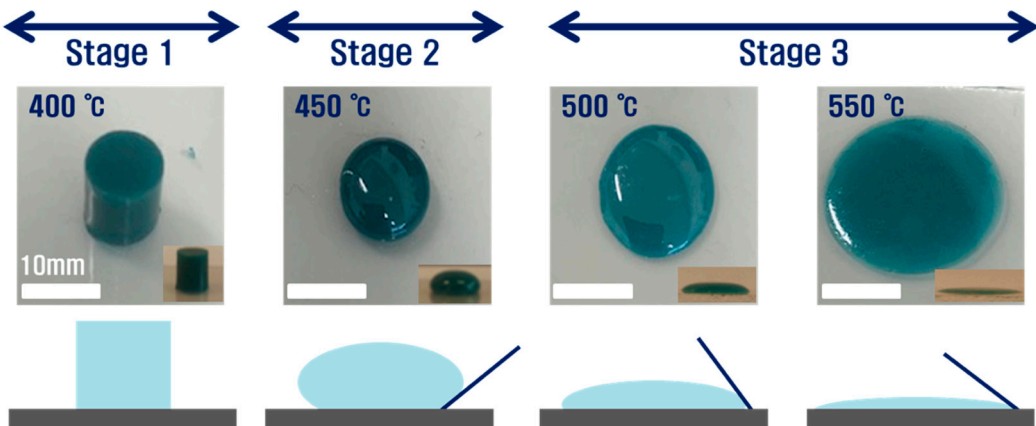

**Figure 6.** The shape change of pellets (1 wt% CuO and 2 wt% Na$_2$CO$_3$ codoped PSAB glass system) as a function of sintering temperature; inset shows the side view of pellets.

Figure 7a shows the XRD spectra for PSAB, 1 wt% CuO doped, and 1 wt% CuO-2 wt% Na$_2$CO$_3$ codoped PSAB glasses. As can be seen from the figure, the broad hump around 2θ ≈ 28° was observed in the XRD pattern, and thus it confirmed the amorphous nature of PSAB-based glass samples are homogeneous regardless of types of dopants. The previous reports confirmed the amorphous nature from XRD in 12 mol% CuO added into the (20 − x)BaO-30ZnO-10Na$_2$O-40P$_2$O$_5$-xCuO glass system [26]. It was reported that the above 14 wt% Na$_2$CO$_3$ provides a nucleation site to improve the crystallization of the glass [23]. Figure 7b shows the XRD patterns of 1 wt% CuO and 2 wt% Na$_2$CO$_3$ codoped PSAB glass system for different sintering temperatures varied from 400 to 550 °C, with an increment of 50 °C. It was observed that the broad hump at an angle (2θ) of 28° was noticed for all the temperatures, indicating the amorphous nature of the PSAB glass system irrespective of the sintered temperature.

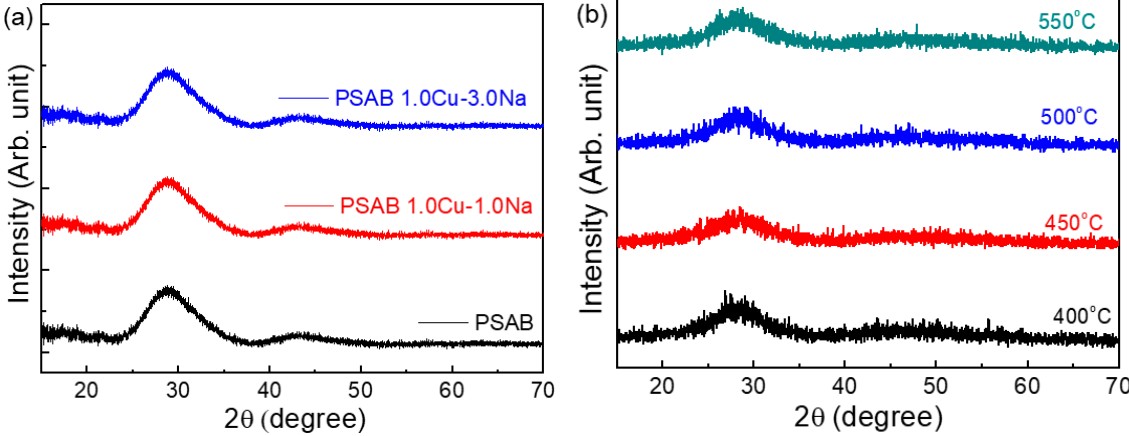

**Figure 7.** (**a**) XRD spectra for base PSAB system, 1 wt% CuO-doped, and 1 wt% CuO and 2 wt% Na$_2$CO$_3$ codoped PSAB glass systems; (**b**) XRD spectra for codoped PSAB glass system sintered at 400–550 °C.

Figure 8 shows the 1 wt% CuO and 2 wt% Na$_2$CO$_3$ codoped (a) PSAB glass ingot, (b) cross-section image of the glass fiber with 180 μm diameter, and (c) glass fiber spool. The glass fiber was pulled using a draw tower. Bronze mold with a diameter of 7 mm and a length of 40 mm was used for the fabrication of glass ingot and fiber. The glass fiber was drawn in the temperature range of 500–535 °C. Three steps are involved in the fabrication of glass fiber. Initially, the temperature of electric furnace increased up to 500 °C with a heating rate of 10 °C/min and then followed this procedure: Seg 1. 500 °C for 60 min → Seg 2. 500 °C for 30 min → Seg 3. 535 °C for 60 min. At the final step, the bob of the preform began to fall, and then the glass fiber was drawn with an outer diameter of 400–550

μm. Finally, the fiber 180 μm diameter was obtained by adjusting the pulling speed with an automatic feedback system. Glass fiber was drawn to a length of 50 m. The fabricated glass fiber could be used as a fiber-type sealing material. Systematic sealing process using a fiber-type sealing material will be explained in the next figure.

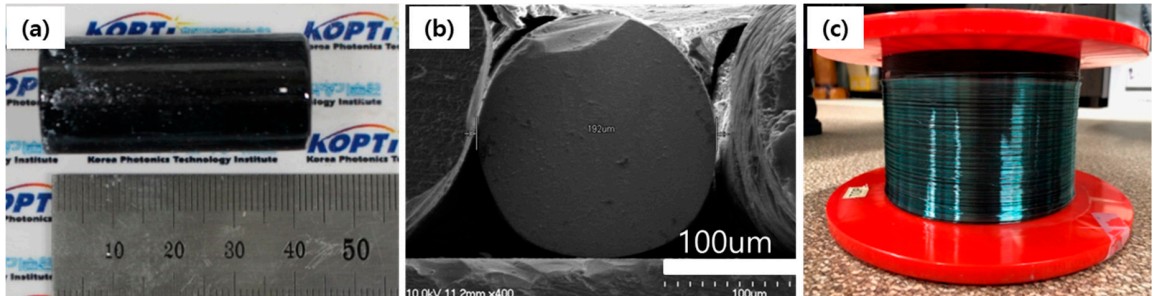

**Figure 8.** (**a**) Glass ingot; (**b**) microscope image of glass fiber with 180 μm diameter; and (**c**) glass fiber spool of 1 wt% CuO-2 wt% $Na_2CO_3$ codoped PSAB system.

In order to optimize the laser sealing process, the microscopic photos of different stages are shown in Figure 9 as a function of sealing cycle and plate temperature. In the reference study [32], Cho et al. reported that screen printed glass frit based on $V_2O_5$ and $TeO_2$ glass had a thickness of ~20 μm and width of 700 μm. It was found that the optimized laser sealing conditions were under laser power of 40 W, laser scan velocity of 300 mm/s, and laser scanning cycle of 200. Compared to the above study, two parameters were modified in the present investigation because the fiber type of sealant had a 180 μm diameter. First, the glass substrates were thermally treated in the temperature range of 130–220 °C for 30 min using hot plates in order to reduce thermal shock after the laser sealing process. Secondly, laser scan velocity was reduced to 40 mm/s after fixing laser power of 35 W. The fiber sealant was not melted at all in the microscopic photos when the glass substrate was below 100 °C, until the laser scanning cycle of 200, and then it looks like delamination. The fiber sealant started to melt when the glass substrate was at a temperature of 130 °C with a laser scanning cycle of 200. However, it has thermal stress cracks. When two FTO glass substrates were heated at 190 °C, the fiber sealant was completely melted at a laser scanning cycle of 200–300, showing crack free surface. The fiber sealant was melted and aggregated showing a sphere type of island of glass molts (pores) regardless of laser scanning cycle of 200–300 after two FTO glass substrates being heated at 220 °C. It was found that two FTO glass substrates using the glass fiber were successfully sealed under laser power of 35 W, laser scan velocity of 40 mm/s, laser scanning cycle of 200–300, and a preheating of 190 °C, resulting crack free surface.

Figure 10a shows a schematic diagram of jig mounted FTO substrates. Figure 10b depicts a universal tensile tester mounted with jig to investigate the laser sealing quality. The strength tests are essential to assess the mechanical resistance of the devices. The laser-sealed FTO substrates with fiber sealant were subjected to mechanical strength using equipment comprising of two stainless steel jigs up and down. Figure 10c shows tensile load changes with time for the laser-sealed FTO substrates using fiber type sealant. It can be observed from Figure 10c that under laser power of 35 W, laser scan velocity of 40 mm/s, laser scanning cycle of 200, and a preheating of 190 °C, uniform relative movement occurred to the fiber sealant and FTO substrates with the testing machine, which had a uniform linear motion, until the breaking force reached the maximum value, at which time the relative displacements of the sealant and FTO substrates were small. The fracture failure mode is ductile fracture, since the fracture of the laser sealing is flush and bright and perpendicular to the direction of the normal stress. The results showed that the maximum load that the samples can withstand is 110 N. This means that the fiber sealant can withstand to a strength of 41 MPa. Sealing stress was not measured on laser-sealed FTO substrates thermally treated at 130 and 220 °C due to defects of delamination and aggregation. It was found that the maximum value of the breaking force was reached

after a certain displacement between upper FTO substrate and lower FTO substrate was formed. In the previous studies, it was reported that sealing strength using screen printed sealant was observed in the range of 12.3–20.5 MPa [3,33]. Compared with values obtained from previous studies above, a sealing strength of 41 MPa seemed to be as strong a candidate as laser sealant.

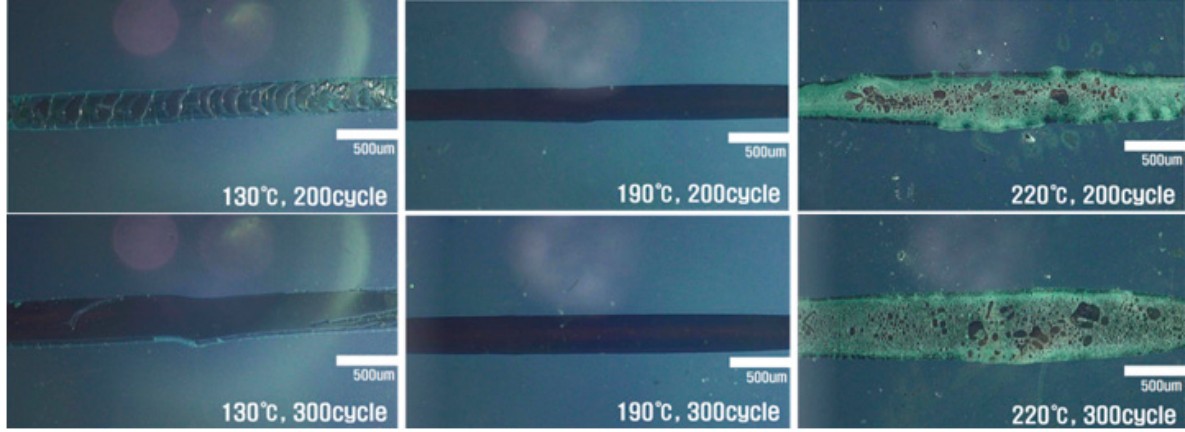

**Figure 9.** The microscopic photos of different stages of laser sealing as a function of sealing cycle and plate temperature. Glass fiber type of sealant is based on 1 wt% CuO-1 wt% $Na_2CO_3$ codoped PSAB glass system.

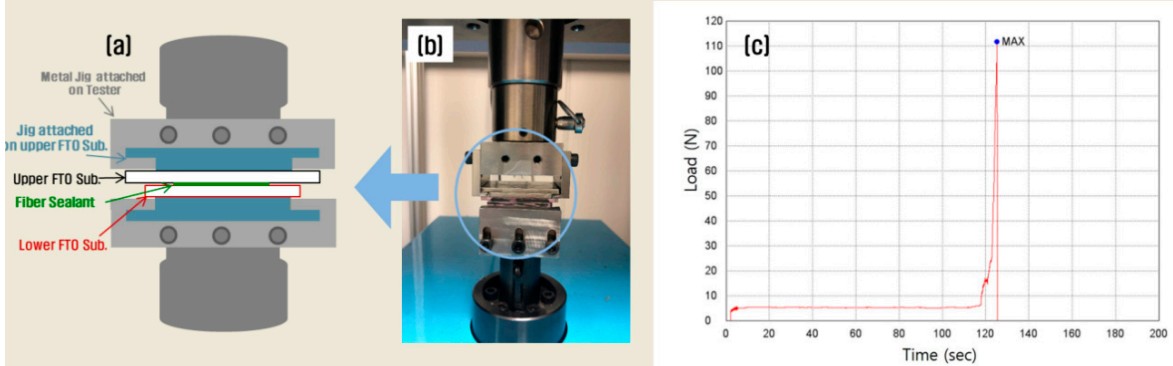

**Figure 10.** (**a**) Schematic diagram of zig mounted with FTO substrates; (**b**) universal tensile tester mounted with zig; (**c**) tensile load changes with time for the laser-sealed FTO substrate (laser power of 35 W, laser scan velocity of 40 mm/s, laser scanning cycle of 300, and preheating of 190 °C).

## 4. Conclusions

We investigated a tuning capability of CuO and $Na_2CO_3$ concentration on thermal properties ($T_g$, $T_x$, $T_s$, CTE) and optical properties of a PSAB glass system. From TMA analysis, the thermal parameters $T_g$ and CTE showed a decreasing trend with increasing CuO concentration. As a function of $Na_2CO_3$ concentration, the $T_g$ decreased but CTE increased. It was observed that the CTE difference for singly CuO- and $Na_2CO_3$-doped PSAB systems was found to be high of the order of $2 \times 10^{-6}$/K. Thus, the tuning capability of codoping on CTE was investigated and the optimum concentrations of CuO and $Na_2CO_3$ were found to be 1 wt% and 2 wt%, respectively. The difference of CTE between the optimized glass system ($9.89 \times 10^{-6}$/K) and the FTO coated glass panel ($10.23 \times 10^{-6}$/K) was found to be $0.34 \times 10^{-6}$/K. The optical absorption at 810 nm also increased for the codoped PSAB system and reached up to 100%. The glass fiber with a diameter of 180 μm was successfully pulled for the codoped PSAB glass system. The tensile strength tests were performed for the laser-sealed FTO substrates, using the glass fiber sealant, and sealing strength was found to be 41 MPa. Therefore, the investigated PSAB-based glass fiber was proved as a potential candidate as a laser sealing material in the packaging industry.

**Author Contributions:** Laser sealing and Analysis, S.Y.K.; Glass Fabrication, J.P.; Formal Analysis, S.H.K.; Writing—Original Draft Preparation, J.H.C.; Writing—Review & Editing, L.K., J.H.K. and J.H.B. All authors have read and agreed to the published version of the manuscript.

**Funding:** This research was funded by [the ministry of trade, industry and energy (MOTIE) and the Ministry of SMEs and Startups (MSS, Korea)] by grant number [No. N0001886, S2725003].

**Conflicts of Interest:** The authors declare no conflict of interest.

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
