# Peer review of "The Tuning Capability of CuO and Na2CO3 Dopant on Physical Properties for Laser Sealing Using Fiber Types of Sealant"

_applsci, doi:10.3390/app10010353_

Round 1

Reviewer 1 Report

No modifications needed. Results were clearly presented.

Author Response

The authors thankful to the reviewer for the positive response of the present form of the manuscript. We updated the manuscript and please find the attachment.

Reviewer 2 Report

An extensive work, with a lot of experiments carefully performed and results presented acurately.

Author Response

(The authors gave the same response as above.)

Reviewer 3 Report

Line 77, the authors mentioned “The glass samples with a dimension of 5*5*8 mm3 were used for the measurement." Please clarify the length, width and height respectively. Line 81, the authors described “The flow button test was performed.” Please provide more details about this test. Line 91, the authors should give the full name of Tg and Tx first, then use the abbreviations. Figure 1 (b), please give the unit for your third axis. Figure 2 (a), the quality of this figure is low, please redraw the figure to make it as clearly as possible, also mark the peak values of glass transition temperature. Figure 6, please add scale bar for all your pictures. Figure 7, please do normalization for your XRD plot and remove all the noisy peaks. Figure 8, please add scale bar for your picture (b). For the conclusion part, it doesn’t have a deep view or discussion about why PSAB based glass fiber has feasibility as a laser sealing materials. need add more mechanism analysis.

Author Response

Point 1. Line 77, the authors mentioned “The glass samples with a dimension of 5*5*8 mm3 were used for the measurement." Please clarify the length, width and height respectively.

Reply. 5*5*8 mm3 was chained into 5(H)*5(W)*8(L) mm3

Point 2. Line 81, the authors described “The flow button test was performed.” Please provide more details about this test.

Reply. The authors thank the reviewer for the suggestions. We have checked the flow button test details and found that the test information was given to the best of author’s knowledge. (See line 229 to 249)

Point 3. Line 91, the authors should give the full name of Tg and Tx first, then use the abbreviations.

Reply. As per the reviewer suggestion, the full name of thermal properties, Tg and Tx was given in the revised manuscript.

Point 4. Figure 1 (b), please give the unit for your third axis.

Reply. As per the reviewer suggestion, the modified Figure 1(b) was given in the revised manuscript.

Point 5. Figure 2 (a), the quality of this figure is low, please redraw the figure to make it as clearly as possible, also mark the peak values of glass transition temperature.

Reply. As per the reviewer suggestion, the Figure 2 (a) was improved and given in the revised manuscript.

Point 6. Figure 6, please add scale bar for all your pictures.

Reply. The modified Figure 6 was given in the revised manuscript.

Point 7. Figure 7, please do normalization for your XRD plot and remove all the noisy peaks.

Reply. The Figure 7 (a) was remeasured and given in the revised manuscript.

Point 8. Figure 8, please add scale bar for your picture (b).

Reply. The Figure 8(b) was revised with scale bar.

Point 9. For the conclusion part, it doesn’t have a deep view or discussion about why PSAB based glass fiber has feasibility as a laser sealing material. need add more mechanism analysis.

Reply. The authors thankful to the reviewer for the deep insight of the manuscript. We have studied the thermal, thermo-mechanical and optical properties of PSAB glasses modified with different concentrations of CuO and Na2CO3. Then, the optimized PSAB glass was drawn into a fiber and its tensile strength was studied. Based on the results, the PSAB glass fiber is suitable as a laser sealing material.
